# Dynamic Response and Energy Absorption Characteristics of a Three-Dimensional Re-Entrant Honeycomb

**Jun Zhang, Boqiang Shi * and Tian Han**

School of Mechanical Engineering, University of Science and Technology Beijing, Beijing 100083, China
* Correspondence: shiboqiang@ustb.edu.cn

**Abstract:** In this paper, we design a new three-dimensional honeycomb with a negative Poisson's ratio. A honeycomb cell was first designed by out-of-plane stretching a re-entrant honeycomb and the honeycomb is built by spatially combining the cells. The in-plane response and energy absorption characteristics of the honeycomb are studied through the finite element method (FEM). Some important characteristics are studied and listed as follows: (1) The effects of cell angle and impact velocity on the dynamic response are tested. The results show that the honeycomb exhibits an obvious negative Poisson's ratio and unique platform stress enhancement effect under the conditions of low and medium velocity. An obvious necking phenomenon appears when the cell angle parameter is 75°. (2) Based on the one-dimensional shock wave theory, the empirical formula of the platform stress is proposed to predict the dynamic bearing capacity of the honeycomb. (3) The energy absorption in different conditions are investigated. Results show that as the impact velocity increases, the energy absorption efficiency gradually decreases. In addition, with the increase of cell angle, the energy absorption efficiency is gradually improved. The above study shows that the honeycomb has good potential in using in vehicle industry as an energy absorption material. It also provides a new strategy for multi-objective optimization of mechanical structure design.

**Keywords:** negative Poisson's ratio; impact response; deformation mode; energy absorption; platform stress

## 1. Introduction

Re-entrant honeycombs have attracted considerable attention due to their excellent mechanical properties, including high stiffness and specific strength [1,2] and superior heat dissipation capabilities [3]. Because of their distinctive energy absorption abilities, they are associated with lightweight material [4–6]. Thus they are extensively used in the field of transportation [7], aerospace and construction [8]. Many studies have recently been published that seek to investigate the in-plane compressing properties [9,10] and out-of-plane compressing properties [2,11] of honeycombs.

Meanwhile, some scholars have used experimental, numerical and theoretical methods [12] to research the compression properties of the re-entrant honeycombs [13–16]. They found that adding ribs in the cell of re-entrant honeycomb can improve Young's modulus and the energy absorption capacity [17,18]. Some researchers have investigated the deformation modes under different compress velocities and found that the re-entrant honeycomb has greater impact resistance than hexagonal honeycomb [5]. Some researchers also introduce a hierarchy into re-entrant honeycombs to investigate the in-plane crashworthiness performance [19]. In recent years, gradient honeycomb has also attracted the attention of many scholars. Studies have found that this honeycomb has better ability by changing the parameters to enhance energy absorption capacity [20–22]. There are many studies on the cell configuration of re-entrant honeycomb and the cell configuration at different compression speeds. Most re-entrant honeycomb is designed by directly stretching 2D configuration to 3D configuration and such design commonly has limited abilities.

In this study, a three-dimensional re-entrant honeycomb is proposed and its in-plane compress performance is investigated. The deformation of the honeycomb under different conditions is calculated by the finite element method, and then the stress formula of the platform is fitted according to the stress–strain curve. Finally, the energy absorption abilities of the honeycomb under different conditions are discussed.

## 2. Finite Element Model and Parameters

### 2.1. Digital Model

The new three-dimensional re-entrant honeycomb is formed by the rotation shown in Figure 1. The cell of traditional re-entrant honeycomb in an out-of-plane tensile is shown in Figure 1a, the representative structural cell (RSC) of the three-dimensional re-entrant honeycomb in orthogonal space is shown in Figure 1b and the array of the re-entrant honeycomb in orthogonal space is shown in Figure 1c. The dimensions of the RSC are shown in Figure 2, where the length of upper and lower cell walls of the re-entrant honeycomb structure is 2L, the length of the ligament connecting the adjacent honeycomb structure is L, t represents the thickness of cell wall of the honeycomb, d represents the width outside the cell wall and α represents the angle between the oblique edge of the re-entrant honeycomb and the horizontal plane (i.e., the cell angle). For the traditional re-entrant honeycomb, the range of α is from 0° to 90°. The internal edges of the traditional re-entrant honeycomb structure are overlapped when α is 0°, and when α is 90°, the re-entrant honeycomb structure becomes a square honeycomb structure.

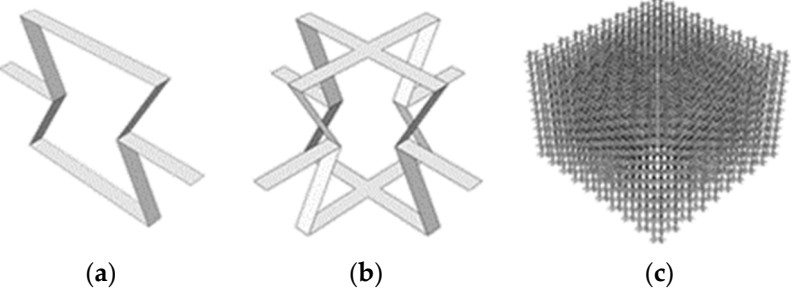

(**a**)         (**b**)         (**c**)

**Figure 1.** Evolution of three-dimensional re-entrant honeycomb structure. (**a**) Traditional re-entrant honeycomb; (**b**) Representative structural cell (RSC); (**c**) Array of the RSC.

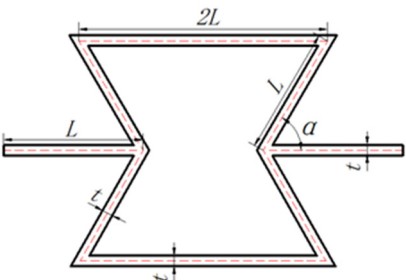

**Figure 2.** Structural parameters of the traditional re-entrant honeycomb (*2L* is the length of upper and lower cell walls, *L* is the length of the ligament, *α* is the angle between the cell walls and *t* is the thickness of cell wall.).

According to the theory of porous materials [1], the relative density of honeycomb materials can be calculated by the ratio of the volume of RSC in Figure 1b to the total volume of three-dimensional space. Therefore, the relative density of three-dimensional honeycomb materials, Δρ, can be written as Formula (1)

$$\Delta\rho = \frac{V_{RVE}}{V_{Total}} = \frac{td(16L - d)}{8(L\sin\alpha + \frac{t}{2})(2L - L\cos\alpha)^2} \tag{1}$$

where $V_{RVE}$ is the volume of three-dimensional honeycomb structure, and $V_{Total}$ is the total volume of representative structural cells in three-dimensional space in Formula (1).

### 2.2. Model Parameters and Constraints

The schematic diagram of the three-dimensional honeycomb calculation model is illustrated in Figure 3, where the direction setting in the simulation model is shown. The specimen formed by the three-dimensional honeycomb is placed between the upper and lower plate. In the test, the lower plate is fixed and the upper plate is driven by the external velocity, and it compresses the honeycomb specimen along the negative direction of the *y*-axis. The cell wall length (L) is 5 mm, the thickness (t) is 0.3 mm and the cell wall width (d) is 1 mm. By changing the cell angle and impact velocity, the dynamic response characteristics of the model in *y*-axis direction are calculated by Abaqus/Explicit dynamic finite element method. The matrix material of honeycomb is selected to be aluminum, assuming that the material is an ideal elastic–plastic material model, which conforms to the Mises yield criterion. The material parameters are given in Table 1. In order to facilitate the calculation of the simulation model, the upper compressed and the lower fixed plate are both regarded as a rigid plate.

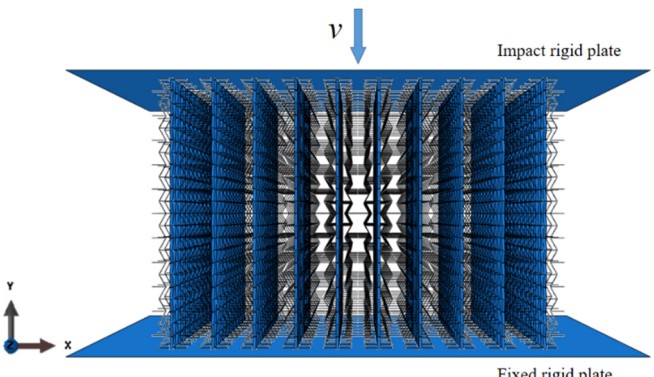

**Figure 3.** The model of simulation.

**Table 1.** Mechanical properties of the aluminum.

| Material | $\rho/(Kg{\cdot}m^{-3})$ | E/GPa | $\sigma_s$/MPa | v |
|---|---|---|---|---|
| Aluminum | 2700 | 69 | 76 | 0.3 |

The Abaqus software is used. In order to ensure the convergence of the calculation process, each cell wall of the three-dimensional honeycomb is discretized by S4R shell element, and five integral points are taken along the direction of cell wall thickness. The general contact of each element of the simulation model is set as an automatic contact and rigid plate, and this can greatly reduce the workload of defining different contact pairs. Therefore, multiple contacts are defined in the calculation, such as the general contact between the rigid plates and the honeycomb specimen and the self-contact between the internal elements of the specimen. Since the contact surfaces cannot be completely smooth, the friction coefficient is set to 0.02 for calculation accuracy [23]. In order to verify the model accuracy, we set simulation parameters as the same as the ones in article [24] and make a comparison. The results show that the deformation is completely consistent in Figure 4, which proves the simulation model is correct. In addition, this article also compares the performances in convergence with different mesh sizes. The force over calculation time is shown in Figure 5. Considering the calculation time and the accuracy of the results, the mesh size is set to 0.5.

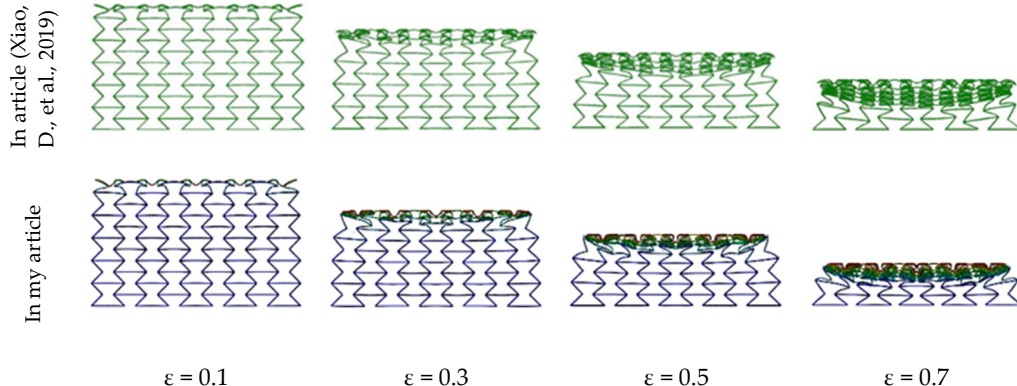

$\varepsilon = 0.1$       $\varepsilon = 0.3$       $\varepsilon = 0.5$       $\varepsilon = 0.7$

**Figure 4.** Comparison of deformation at speed of 100 m/s. Reprinted with permission from ref. [24]. Copyright 2022 Elsevier.

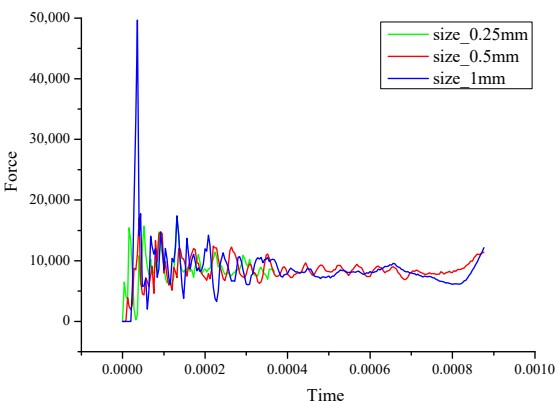

**Figure 5.** Comparison the results of different mesh sizes.

According to the calculation results in [23], the dynamic response of honeycomb specimens can be stabilized when the number of cells in each axial direction exceed 10. Therefore, the specimen has 10 structural cells in the *x*-axis and *z*-axis directions, and the number of cells in the *y*-axis direction change according to the cell angle. The number of cells set in each axis direction are marked in Figure 6. In order to facilitate the comparison of energy absorption efficiency, the height of specimen in *y*-axis is maintained at about 95 mm.

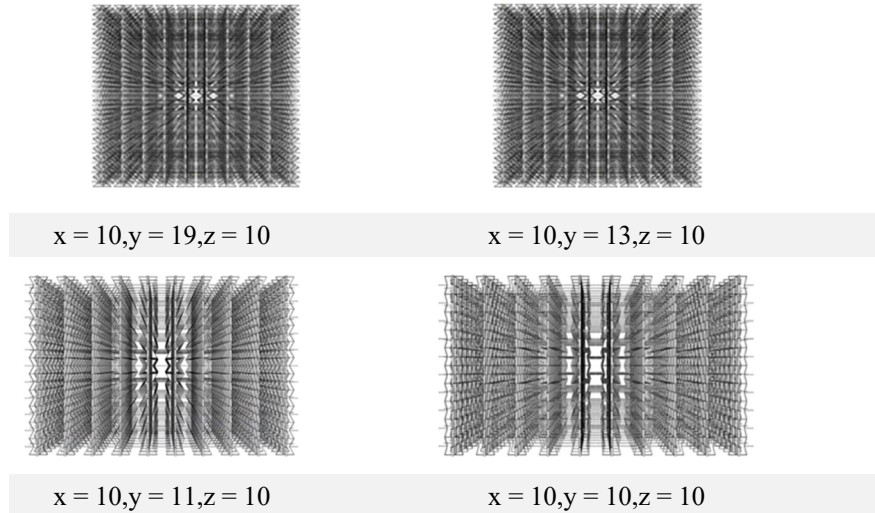

x = 10,y = 19,z = 10       x = 10,y = 13,z = 10

x = 10,y = 11,z = 10       x = 10,y = 10,z = 10

**Figure 6.** Number of different structural cells.

*2.3. Calculation Critical Velocity*

The impact velocity is a significant index affecting the dynamic response characteristics of materials. The dynamic response process is discussed under three conditions:

1.  When the impact velocity is lower than the first critical impact velocity (i.e., the notch wave velocity), the whole specimen is slowly compressed, and the force is relatively uniform during the impact process. The material undergoes quasi-static deformation.
2.  When the impact velocity exceeds the first critical impact velocity (i.e., notch wave velocity), the impact process transits from the overall deformation to the local deformation and the local deformation band is formed. With the increase of impact velocity, the local deformation of the upper end of the specimen is more obvious.
3.  When the impact velocity is higher than the second critical impact velocity, the local deformation zone will propagate from the upper end to the lower end of the specimen in the mode of a shock wave.

The impact velocity of honeycomb material with local deformation zone during impact is called the first critical impact velocity, and its calculation formula is as follows:

$$v_{c1} = \int_0^{\varepsilon_1} \sqrt{\frac{\sigma(\varepsilon)}{\Delta\rho\rho_A}} d\varepsilon \tag{2}$$

In Formula (2), $\varepsilon_1$ is defined as the corresponding nominal strain (i.e., initial strain) when the stress reaches the stress peak for the first time in the process of impact fluctuation. $\sigma(\varepsilon)$ represents the elastic modulus of honeycomb material in the online elastic stage, and $\Delta\rho$ is the relative density of honeycomb material. $\rho_A$ is the density of the honeycomb material.

The shock velocity when honeycomb material deformation is compressed with shock wave deformation characteristics is known as the second critical shock velocity. The calculation formula is as follows:

$$v_{c2} = \sqrt{\frac{2\sigma_p\varepsilon_3}{\Delta\rho\rho_A}} \tag{3}$$

$\sigma_p$ is the plateau stress of honeycomb materials under quasi-static compression in Formula (3), and $\varepsilon_3$ is the locking strain, that is, the strain value at the beginning of the densification stage of honeycomb materials.

According to the above Formulas (2) and (3), when the cell parameters are as follows: t = 0.3 mm, d = 1 mm, $\alpha$ = 45° and L = 5 mm, the first critical impact velocity $V_{c1} \approx 11$ m/s and the second critical impact velocity $V_{r2} \approx 62$ m/s are calculated. This paper selects the impact velocity $V_1 = 3$ m/s ($V_1 < V_{cr1}$), $V_2 = 20$ m/s ($V_{cr1} < V_2 < V_{cr2}$) and $V_3 = 200$ m/s ($V_{cr2} < V_3$) to study the impact deformation in order to observe the influence of different impact velocities on the dynamic response of three-dimensional re-entrant honeycombs.

## 3. The Result of Simulation and Discussion

*3.1. Deformation Mode*

Under different impact velocities, the deformation of model is an important characteristic of the dynamic response of honeycomb. The reason for the deformation is that the wall of cell inside the specimen is rotation and buckling under external loads.

When the angle $\alpha$ is 45°, the deformation of the three-dimensional honeycomb under three different impact velocities of low speed ($V_1 = 3$ m/s), medium speed ($V_2 = 20$ m/s) and high speed ($V_3 = 200$ m/s) are shown in Figures 7–9, respectively. The nominal strain ($\varepsilon$) in the Figures is the ratio of the displacement of the specimen in *y*-axis direction to the initial height.

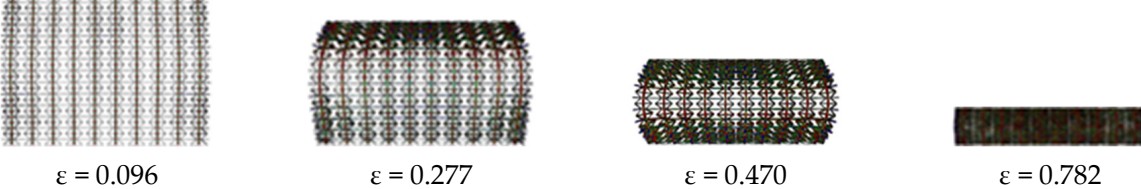

| $\varepsilon = 0.096$ | $\varepsilon = 0.277$ | $\varepsilon = 0.470$ | $\varepsilon = 0.782$ |
|---|---|---|---|

**Figure 7.** Deformation of specimen at low speed (3 m/s and $\alpha = 45°$).

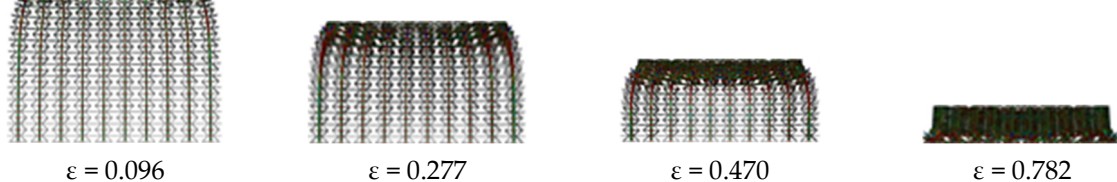

| $\varepsilon = 0.096$ | $\varepsilon = 0.277$ | $\varepsilon = 0.470$ | $\varepsilon = 0.782$ |
|---|---|---|---|

**Figure 8.** Deformation of specimen at medium speed (20 m/s and $\alpha = 45°$).

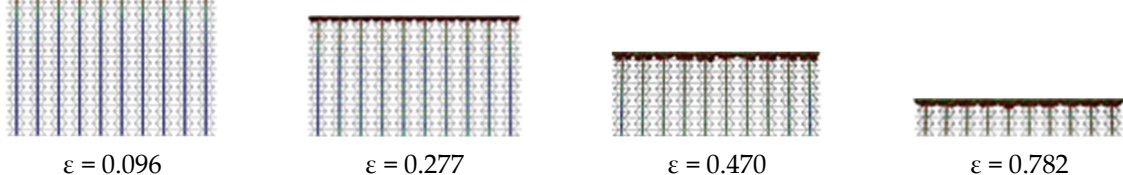

| $\varepsilon = 0.096$ | $\varepsilon = 0.277$ | $\varepsilon = 0.470$ | $\varepsilon = 0.782$ |
|---|---|---|---|

**Figure 9.** Deformation of specimen at high speed (200 m/s and $\alpha = 45°$).

In the case of low velocity ($V_1$ = 3 m/s), the deformation process of three-dimensional honeycomb can be roughly divided into four stages. Phase I ($\varepsilon$ = 0.096) is mainly the rotation of the inclined cell wall inside the three-dimensional honeycomb. The results show that the more the impact velocity is close to the velocity of quasi-static compression, the more uniform the stress is in the compression process. The specimen is uniformly deformed all the time in this stage and the upper and lower ends are close to the middle under the action of the *x*-axis force generated by the rotation of the cell wall, which shows that the specimen has a specific negative Poisson's ratio. The middle part of the specimen in the *y*-axis direction has little force and almost no deformation, so the middle part of the specimen is convex during compression. In Phase II ($\varepsilon$ = 0.277), the deformation is mainly caused by the continuous rotation of the inclined cell wall in the upper structure of the specimen in order to withstand the compression deformation in the *y*-axis direction. Therefore, under the action of the cohesion in the *x*-axis direction, the upper end of the specimen has obvious concave phenomenon. When the upper end is compressed to a certain extent, it enters Phase III ($\varepsilon$ = 0.470). The internal inclined cell wall of the upper end of the specimen will maintain a certain angle in the process. The pressure is transferred from the upper end of the specimen to the lower end, which leads to the rotation of the internal inclined cell wall of the lower end of the specimen. The cohesive force begins to contract to the middle, and the concave shape appears to bear the impact force transmitted from the upper end. When the upper and lower ends are basically symmetrical and the upper and lower ends of the specimen are concave and the middle part is convex, forming a 'barreling' state, this stage is completed. Then the deformation enters Phase IV ($\varepsilon$ = 0.782), the specimen continues compressing in the *y*-axis direction. Because the upper and lower ends of the specimen have been compressed to a certain extent, the inclined cell wall of the middle part of the specimen in the *y*-axis direction will rotate. Under the action of transverse force in the *x*-axis direction, the inclined cell wall begins to converge to the middle vertical plane. When the inner cell wall of the structure basically parallel, the adjacent cell walls reach full contact density, and the compression is stopped.

In the case of medium velocity ($V_2$ = 20 m/s), the deformation can be divided into two stages. The first stage includes the deformations where the strain is between 0 to 0.47. When $\varepsilon$ = 0.096, the impact energy cannot be transmitted to the lower end of the specimen, resulting in the compression of the upper end of the specimen. The inclined cell wall begins to rotate inside the specimen. Because the impact velocity is higher than the one at low speed, the horizontal cell wall will produce buckling and the external impact energy is absorbed. Due to the rotation and buckling of the inner cell wall of the specimen structure, the upper part of the specimen will produce an obvious concave deformation, where negative Poisson's ratio characteristics are shown. When $\varepsilon$ = 0.277 and $\varepsilon$ = 0.470, the deformation still presents in the first stage. The specimen is compressed layer by layer from top to bottom. In the second stage ($\varepsilon$ = 0.782), the compression of the specimen is passed layer by layer to the lower of the specimen, and the internal cell walls begin to contact each other and produce dense compression. At this stage, the bottom layer of the specimen cannot bear the force in the $x$-axis direction due to the lessened friction force, so there is a slight rollover phenomenon.

At high speeds ($V_3$ = 200 m/s), the compression deformation can be seen as one stage. The inertia effect plays a leading role due to the fast impact speed. The inclined cell wall in the specimen structure cannot produce rotation and only buckling deformation occurs. The specimen is compressed layer by layer from upper to lower, until the compression reaches to the bottom and the inner cell wall of the structure is fully contacted and dense. Negative Poisson's ratio can be hardly shown in this process.

It can be concluded that when the speed is low, the whole specimen is more evenly deformed from top to bottom. Due to the effect of friction, when the speed is low, the specimen presents 'barreling'. With the speed increases, this phenomenon gradually disappears, inertial force plays a major role, the specimen deforms from the upper end and the bottom deformation becomes smaller.

The above research discusses the influence of different impact velocities on the deformation of specimen at the same cell angle. The deformations of specimen with different cell angles under the same impact velocity is discussed next. In the test, the cell wall length L = 5 mm is unchanged, the impact velocity is 3 m/s and the compression deformation is set as $\varepsilon$ = 0.186. The deformation of specimens with different cell angles $\alpha$ = 30°, 45°, 60° and 70° are shown in Figure 10a–d, respectively.

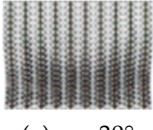 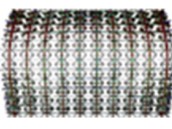 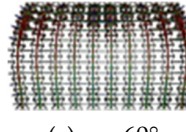 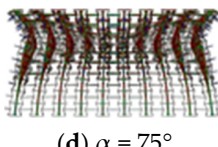

(**a**) $\alpha$ = 30°                    (**b**) $\alpha$ = 45°                    (**c**) $\alpha$ = 60°                    (**d**) $\alpha$ = 75°

**Figure 10.** Deformation at different angles at speed of 3 m/s ($\varepsilon$ = 0.186).

When the cell angle $\alpha$ = 30°, the impact energy is first transferred from the upper of the specimen to the lower. Shrinkage deformations in the vertical direction are produced in the middle part. The shrinkage deformation of the lower part of the specimen is greater than that of the upper part of the specimen. The deformation mode of this case is that the deformation is small near the upper end and the deformation is great next to the lower end.

When the cell angle is $\alpha$ = 45°, the impact energy is also transferred from the upper to the lower of the specimen. In the process of the compression deformation of the specimen, due to the rotation of the inclined cell wall inside the specimen, the transverse deformation in the $x$-axis direction is produced. Both ends of the specimen shrink and the middle position of the $y$-axis direction is relatively small, so it finally presents the 'barreling' mode.

When the cell angle is $\alpha$ = 60°, the impact energy is transferred from the upper to the lower parts of the specimen. In the process of compression deformation, the situation is basically the same as that of $\alpha$ = 45°, so it finally presents the 'barreling' state. Compared with $\alpha$ = 45°, the deformation at the lower part of the specimen is smaller, and the deformation is mainly concentrated in the upper part of the specimen.

When the cell angle $\alpha = 75°$, the internal cell wall of the specimen is sparser due to the larger angle of $\alpha$. When the impact begins from the upper part of the specimen, it shows that the upper part of the specimen is prone to deformation, while the lower part does not produce deformation. A specific position at the upper part of the specimen shows the necking phenomenon, and the specimen has obvious negative Poisson's ratio characteristics.

By studying the impact deformation of different cell angles under the same impact velocity, it is found that when the cell angle $\alpha = 30°$, the deformation of the negative Poisson's ratio mainly occurs at the lower part of the specimen, and the deformation morphology is different from that with other angles where the lower shrinkage is larger than the upper shrinkage. When the cell angle is $\alpha = 45°$ and $\alpha = 60°$, the deformation modes are basically the same, showing the shape of 'barreling'. Besides the phenomenon of negative Poisson's ratio still exists at the upper and lower ends of the specimen. However, with the increase of the cell angle, the deformation at the lower part of the specimen gradually weakens and mainly concentrates on the upper part of the specimen. When the cell angle is $\alpha = 75°$, the upper end of the specimen displays an obvious 'necking' phenomenon, and the lower end almost has no deformation. It can be concluded that under the same impact velocity, with the increase of the cell angle, the deformation position gradually transits from the lower of the specimen to the upper end and shows different deformation modes.

When the impact velocity is low ($V_1 = 3$ m/s), the nominal stress–strain curve of the specimen is shown in Figure 11. When the three-dimensional honeycomb parameters are $\alpha = 45°$, $L = 5$ mm and $t = 0.3$ mm, the $\varepsilon$ represents the nominal strain in the horizontal coordinate, that is, the ratio of the compression reaction of the upper rigid plate to the initial contact area of the specimen, and the $\sigma$ represents the nominal stress in the vertical coordinate.

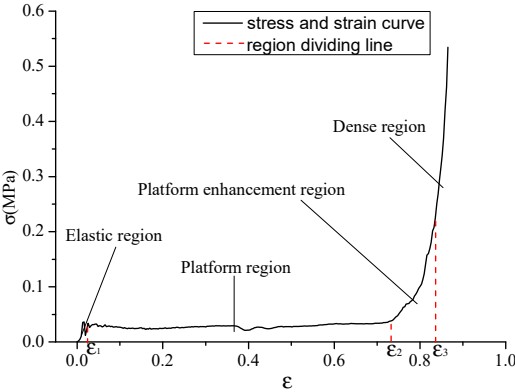

**Figure 11.** Nominal stress–strain curve of specimen under in-plane impact.

When honeycomb is subjected to in-plane compression, it is studied according to [25]. The compression process of traditional honeycomb is generally divided into three regions, as shown in Figure 11, which are the linear elastic region, platform region and dense region. However, compared with the traditional honeycomb, the compression process of the three-dimensional re-entrant honeycomb is divided into four regions, namely, the linear elastic region, platform region, platform enhancement region and dense region.

The linear elastic region is a process, where the compression stress of the upper rigid plate suddenly increases and reaches the initial stress peak in a very short time. After that ($\varepsilon = \varepsilon_1$), the stress begins to fluctuate and finally tends to be stable. In the platform region, the compressive stress of the specimen after reaching the initial strain $\varepsilon_1$ fluctuates around a certain value and remains relatively stable. The specimen undergoes great compressive deformation in this stage, so it is the main stage of energy absorption. After the end of the platform stage, it enters the platform enhancement region. With the continuous increase of the compressive strain of the specimen, the stress no longer remains relatively stable,

but gradually increases with a specific slope and exceeds the platform stress value to a certain extent. After the strain at the end of the enhancement stage reaches $\varepsilon_3$, the cells in the specimen begin to contact with each other in dense region. At this region, the stress value of the specimen rises sharply in a small strain stage until the inner wall of the cells in the specimen is completely bonded together and the dense stage ends.

### 3.2. Platform Stress

When the stress remains in a relatively stable region from $\varepsilon_1$ to $\varepsilon_2$, the stress in this region is called the platform stress ($\sigma_p$). It is an important indicator for describing the dynamic response characteristics of the honeycomb and can be calculated by the following Formulas (4) and (5):

$$\sigma_P = \frac{\int_{\varepsilon_1}^{\varepsilon_3} \sigma(\varepsilon) d(\varepsilon)}{\varepsilon_1 - \varepsilon_3} \tag{4}$$

$$\sigma(\varepsilon) = \frac{F(\varepsilon)}{L_x \times L_z} \tag{5}$$

In Formula (4), $\varepsilon_1$ is the initial strain, that is, the corresponding strain value when the initial stress is just stable and reaches the platform stress, so the value of $\varepsilon_1$ is very small. In order to achieve a high calculation accuracy, the value of $\varepsilon_1$ in this paper is set as 0.013. $\varepsilon_3$ is dense strain, that is, the strain corresponding to the contact between adjacent cell walls within the specimen. In Formula (5), the value of $F(\varepsilon)$ is derived from the average value of the force of the upper rigid plate in the platform area obtained by the simulation. $L_x$ is the length of the specimen in the $x$-axis direction, and $L_z$ is the length of the specimen in the $z$-axis direction.

According to the one-dimensional shock wave theory [1,25], the formula of platform stress is obtained as follows:

$$\sigma_P = m\sigma_s\Delta\rho^2 + \frac{\Delta\rho\rho_s v^2}{1 - n\Delta\rho} \tag{6}$$

where $\sigma_s$ represents the yield stress of the matrix material, $\Delta\rho$ represents the relative density of the designed honeycomb material, $\rho_s$ represents the density of the matrix material and $m$ and $n$ are the coefficients to be calculated or fitted.

The stress over different velocities are calculated using the FEM method and listed in Table 2. According to Formula (6) and the data points in Table 2, three curves are fitted and plotted in Figure 12, and the formula of the three curves are obtained by linear regression, so as to solve the parameters $m$ and $n$ in the formula. By verifying the results, it has been found that the value of $n$ is too large, mainly because the value of $\rho$s in the formula leads to the inapplicability of the formula. It is found that the value of $n$ conforms to the linear distribution by observation, so the calculation formula of platform stress is modified by fitting the value of $n$ again. The modified formula is as follows:

$$\sigma_p = 2.188\sigma_s\Delta\rho^2 + \left(0.0023\Delta\rho + 6.948e^{-9}\right)v^2 \tag{7}$$

The comparison between the results of the three-dimensional honeycomb platform stress under different densities by FEM and the curves of the formula are shown in the Figure 12 as well. It can be seen that the fitting of the result is better, and the smaller the relative density is, the higher the fitting degree is. Therefore, the rationality of the correction formula is verified.

**Table 2.** Platform stress in different condition.

| v/(m/s) | σp/(MPa) | | | |
|---|---|---|---|---|
| | Δρ = 0.037 | Δρ = 0.02 | Δρ = 0.012 | Δρ = 0.008 |
| 3 | 0.037 | 0.026 | 0.023 | 0.019 |
| 7 | 0.038 | 0.031 | 0.028 | 0.022 |
| 20 | 0.081 | 0.053 | 0.039 | 0.035 |
| 35 | 0.167 | 0.109 | 0.058 | 0.055 |
| 70 | 0.623 | 0.303 | 0.192 | 0.171 |
| 100 | 1.450 | 0.615 | 0.378 | 0.314 |
| 200 | 3.778 | 2.158 | 1.403 | 1.079 |

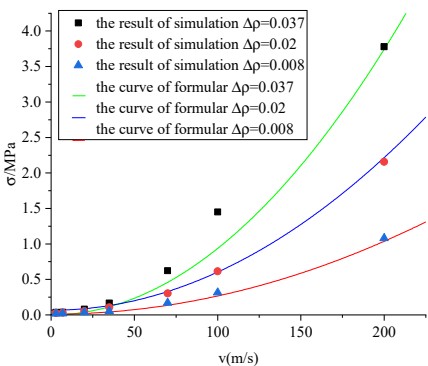

**Figure 12.** Comparison of accuracy under different densities.

### 3.3. Energy Absorption

The effects of the impact velocity and cell angle on the stress and strain of specimen during impact are studied below. Firstly, under the condition of the constant cell angle (relative density), the nominal stress and strain curves of three-dimensional re-entrant honeycomb are obtained by simulation. It can be concluded from Figure 13a that under this condition, the stress increases with the increase of velocity. In addition, the stress and strain of different cell angles (different densities) under the same impact velocity can be obtained in Figure 13b. It can be seen from Figure 13b, the stress decreases with the increase of the cell structure angle.

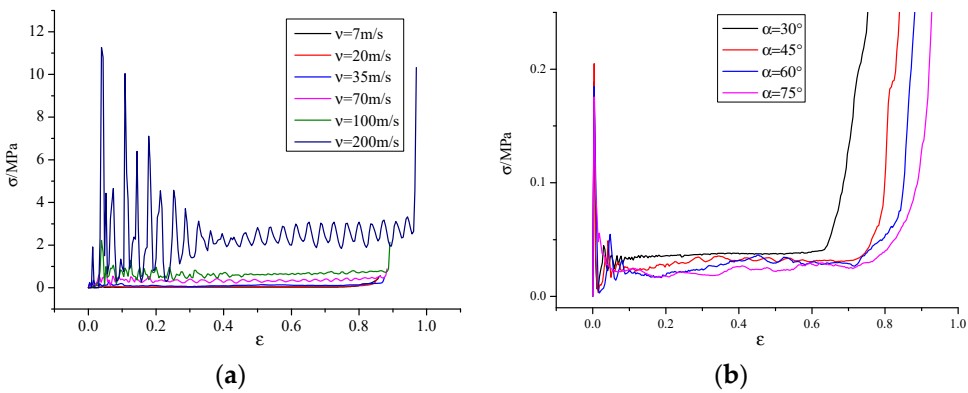

**Figure 13.** The relationship between nominal stress and strain of specimen. (**a**) The stress and strain of different velocities. (**b**) The stress and strain of different cell angles.

Energy follows the first principle of thermodynamics under external loads, which can be expressed by Formula (8). Ignoring the energy of friction loss and the energy of the damping dissipation of surrounding media in Formula (8), the external work is mainly

converted into kinetic energy and the internal energy absorbed by the impact object, so the sum of the two is regarded as the total energy absorbed by the material.

$$E_w + E_{qb} = E_u + E_k + E_f \tag{8}$$

where $E_u$ is the internal energy of the material, $E_k$ is the kinetic energy of the material, $E_f$ is the energy of the contact friction loss, $E_w$ is the work done by the external load and $E_{qb}$ is the energy dissipated by the surrounding medium damping.

The curve of total energy over strain is shown in Figure 14. Figure 14a shows that when the cell angle (relative density) unchanged, the ability to absorb energy during compression increases with the increase of velocity. In addition, when the velocity is constant ($V_1 = 3$ m/s), the ability to absorb energy increases with the increase of cell angle shown Figure 14b. Therefore, the energy absorption ability of three-dimensional honeycomb can be improved by changing the impact velocity and cell angle.

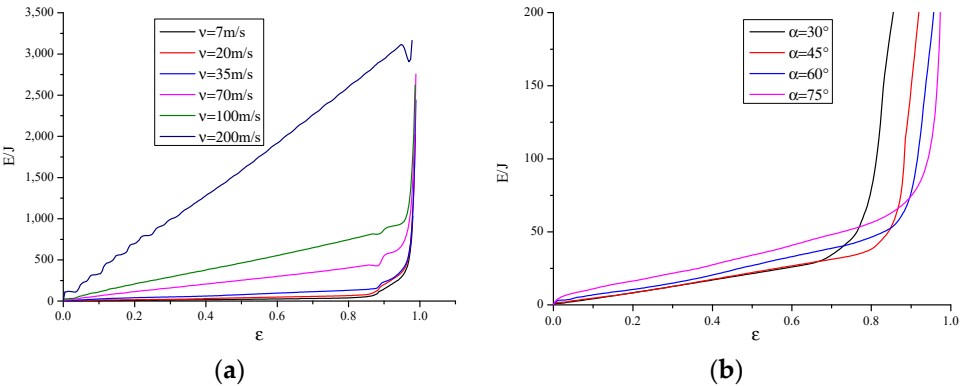

(**a**)            (**b**)

**Figure 14.** Relationship between energy absorption and strain of specimens. (**a**) E (total energy) with different velocities. (**b**) E (total energy) with different angles.

In order to investigate the energy absorption distribution of three-dimensional honeycomb structure under in-plane impact, the internal energy distribution coefficient Φ (the proportion of internal energy in total absorbed energy) is defined. The formula is as follows:

$$\Phi = \frac{Eu}{Ek + Eu} \tag{9}$$

The influence of impact velocity and cell angle on the internal energy distribution coefficient Φ during the impact of specimen is studied below. The variation of the internal energy distribution coefficient Φ with the nominal strain is shown in Figure 15. The condition that relative density (cell angle) of the honeycomb is constant and the impact velocity is different in Figure 15a. The result show that the impact velocity has a great influence on the internal energy distribution coefficient Φ. With the increase of the impact velocity, the internal energy distribution coefficient Φ decreases accordingly, and its value gradually decreases from 0.95 at a low speed impact (V = 7 m/s) to 0.45 at a high speed impact (V = 200 m/s). It can be concluded that when the impact velocity is lower than the second critical impact velocity, the honeycomb absorbs most of the internal energy. With the increase of impact velocity, the proportion of internal energy distribution decreases due to the increase of inertial effect. In addition, when the impact velocity (V = 3 m/s) is constant and the relative density (cell angle) changes, the internal energy distribution coefficient Φ also depends on the cell angle, as shown in Figure 15b. Under the same impact velocity, the internal energy distribution coefficient Φ increases slightly with the increase of cell angle. It can be concluded that the effect of impact velocity on the absorption of impact energy is greater than the cell angle.

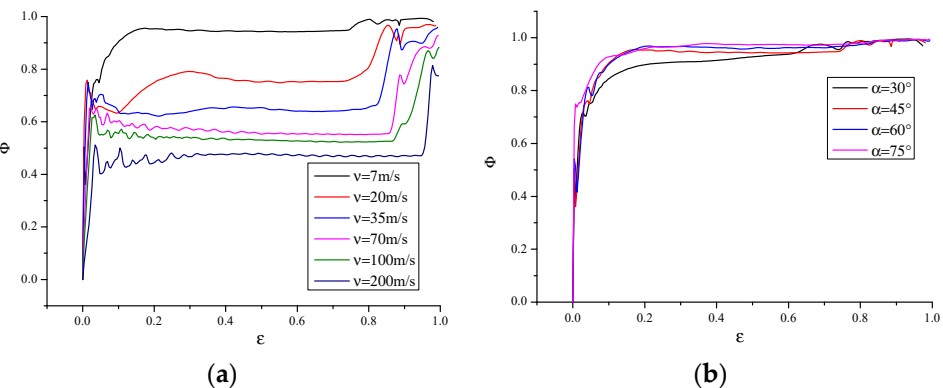

**Figure 15.** Relationship between energy distribution coefficient and strain. (**a**) $\Phi$ (internal energy distribution coefficient) with different velocities. (**b**) $\Phi$ (internal energy distribution coefficient) with different cell angles.

## 4. Conclusions

Based on the traditional re-entrant honeycomb, a 3D honeycomb through a spatial combination is designed in this paper. Then, the dynamic response characteristics of three-dimensional re-entrant honeycomb are numerically analyzed by the explicit dynamic finite element method, and the following conclusions are drawn.

Three-dimensional honeycomb with different cell angles exhibits different deformation modes at the same speed. When the cell angle is 30°, it presents the different compression deformation morphology where the shrinkage of the lower part is larger than that of the upper part. When the cell angles were 45° and 60°, the compressive deformation pattern of 'barreling' was presented. When the cell angle is 75°, the specimen shows the necking phenomenon, which conform to the negative Poisson's ratio material under axial compression. The rotation and bending deformation of the cell wall are the main reasons for the negative Poisson's ratio of the honeycomb. With the increase of impact velocity, the deformation localization is obvious, the inertial effect is gradually enhanced and the negative Poisson's ratio characteristic is weakened. In addition, the stress–strain curve of three-dimensional re-entrant honeycomb adds the platform enhancement region compared with the traditional honeycomb.

Based on the one-dimensional shock wave theory, the empirical formula of the platform stress of three-dimensional re-entrant honeycomb is given, which is proved to be in good agreement with the FEM calculation results. In addition, it can be seen from the fitting results that the smaller the relative density, the higher the fitting degree.

The impact velocity has a great influence on the internal energy distribution coefficient $\Phi$. With the increase of the impact velocity, the internal energy distribution coefficient $\Phi$ accordingly decreases, and its value gradually decreases from 0.95 at low speed (V = 7 m/s) to 0.45 at high speed (V = 200 m/s). Therefore, it can be concluded that when the impact velocity is lower than the second critical impact velocity, the material in this paper mainly absorbs internal energy. With the increase of impact velocity, the proportion of internal energy distribution will decrease due to the increase of the inertial effect.

**Author Contributions:** This research was supervised by B.S. and T.H.; the investigation, software and writing were carried out by J.Z. All authors have read and agreed to the published version of the manuscript.

**Funding:** This work was supported by National Key R&D Program of China (2018YFC0810500).

**Data Availability Statement:** The raw/processed data required to reproduce these findings cannot be shared at this time as the data also forms part of an ongoing study.

**Acknowledgments:** The authors would like to thank the support from the National Key R&D Program of China (2018YFC0810500).

**Conflicts of Interest:** The authors declare no known competing financial interest or personal relationship that could have appeared to influence the work reported in this paper.

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
