# Peer review of "Dynamic Response and Energy Absorption Characteristics of a Three-Dimensional Re-Entrant Honeycomb"

_electronics, doi:10.3390/electronics11172725_

Round 1

Author Response

Thank you very much for your suggestions, I have corrected the questions you asked. Please see the attachment.

Reviewer 2 Report

This paper investigates the dynamic responses of 3D re-entrant honeycomb composites under impact loadings. The effects of impact velocity and internal angle are studied. Energy absorption is treated as a key parameter to evaluate the behavior of different honeycomb under impact. This idea of this paper is clear and the results have been well documented. I recommend accept this paper after minor revisions.

1. Please carefully check the word "re-entrant". In the manuscript, I have seen "reentrant" and "re entrant". It must be consistent.

2. More details can be given to the caption of Figure 1. Like "Evolution of three-dimensional reentrant honeycomb structure: (a) ..., (b) ..., and (c)...

3. In line 87, mises should be Mises.

4. In Section 2.3, it is stated that the impact velocity is a significant index affecting the dynamic response characteristics of materials. This is not sufficiently correct. Impact velocity together with impactor mass determine the impact response. In addition, a classic paper by Olsson can be cited in Section 2.3. Olsson, R., 2000. Mass criterion for wave controlled impact response of composite plates. Composites Part A: Applied Science and Manufacturing31(8), pp.879-887.

5. In line 245, please point out if there are still any negative-Poisson's-ratio effects when alpha is equal to 45 and 60 degree.6. The stage numbers as described for Figures 5-7 can be added to Figures 5-7.

6. In line 308, \delta rho should be relative density, instead of density.

7. Figures 10 - 13 are not very clear. They look like they are from screenshots. The authors should improve the quality of them.

8. Besides deformation plots shown in Figures 5-8, I think strain and stress plots are also helpful.

Author Response

Thank you very much for your suggestions. Please see the attachment and I have corrected using red labels.

Reviewer 3 Report

The paper is well written.  I have only one query.

1. The results are presented for different  impact velocity. Since it is FE solution using ABAQUS.  There is no check for validity of the results and even mesh convergence is not done. If possible, give at least one validation with existing literature. 

Author Response

Thank you very much for your suggestions, I have corrected the question you asked. please see the attachment.

Round 2

Reviewer 1 Report

MDPI - Electronics

Title: Revised version of “Dynamic Response and Energy Absorption Characteristics of A Three-dimensional Re- entrant Honeycomb” by Zhang et al.

It is good to see that the authors have attempted to answer the reviewers’ questions and improve the quality of the manuscript. However, I will have to say that while revising, they have corrected only the language mistakes that I pointed out (as examples) in my first set of comments, and have conveniently ignored all other mistakes. In fact, the manuscript requires a thorough proofreading by a language expert. Even the reply document is full of mistakes and at times I had to guess what the authors were trying to say.

Anyway, I have a few more specific comments:

In Figure 4, the caption in the revised manuscript should give the full information of "The article". The way it has been mentioned is not proper referencing.

The readers cannot anticipate what the authors will be doing in the next article; so, my comments came from a reader’s point of view.

The caption of Figure 5 should mention “mesh sizes” – simply the word, “sizes”, does not make it clear at all.

The selection of frictional co-efficient should be made clear in the manuscript itself.

In the revised manuscript (L98 – L102), the text says “I”; however, it is a multi-authored article and that should be reflected in the text.

Line 252: “…… the shape of ‘()’” is not the right way to define the shape – generally this non-homogeneous deformation is called ‘barreling’.

With all the aforementioned comments, I cannot accept the manuscript in its current form. The authors should devote enough time in revising the text and getting all the language and other silly mistakes corrected properly.

Author Response

Thank you for your suggestions, please also see the attachment.

Thank you very much for your suggestions sincerely. This is my first English article and it certainly have many mistakes. As you said, my letters are also hard to read. Thanks for your reading and giving me many valuable comments two times. Nowadays, I have corrected my manuscript thoroughly. It must have many mistakes, but the ability of English writing improved by your valuable suggestions. Thank you very much!

  1. I have corrected my manuscript thoroughly.

  1. In Figure 4, the caption in the revised manuscript corrected to “In article [24]”

  1. The caption of Figure 5 have corrected to “Comparison the results of different mesh sizes”

  1. The selection of frictional co-efficient should be made clear in the manuscript itself.

Since the contact surfaces (between rigid palate and specimen) cannot be completely smooth, the friction coefficient is set to 0.02 for calculation accuracy [24].

  1. In the revised manuscript (L98 – L102), the text says “I”; however, it is a multi-authored article and that should be reflected in the text.

I have changed the mistake.

  1. Line 252: “…… the shape of ‘()’” is not the right way to define the shape – generally this non-homogeneous deformation is called ‘barreling’.

I have corrected the shape of ‘()’ to ‘barreling’ all my manuscript.
